# The Discovery of Ribosomal Protein bL31 from *Escherichia coli*: A Long Story Revisited

**DOI:** 10.3390/ijms24043445

**Published:** 2023-02-08

**Authors:** Akira Wada, Masami Ueta, Chieko Wada

**Affiliations:** Yoshida Biological Laboratory Inc., Biological Information Research, Yoshida Biological Laboratory, 11-1 Takehanasotoda-cho, Yamashina-ku, Kyoto 607-8081, Japan

**Keywords:** ribosomal protein bL31, ribosome subunit assembly, protease 7, RFHR 2D PAGE method, translational activity

## Abstract

Ribosomal protein bL31 in *Escherichia coli* was initially detected as a short form (62 amino acids) using Kaltschmidt and Wittmann’s two-dimensional polyacrylamide gel electrophoresis (2D PAGE), but the intact form (70 amino acids) was subsequently identified by means of Wada’s improved radical-free and highly reducing (RFHR) 2D PAGE, which was consistent with the analysis of its encoding gene *rpmE*. Ribosomes routinely prepared from the K12 wild-type strain contained both forms of bL31. Δ*ompT* cells, which lack protease 7, only contained intact bL31, suggesting that protease 7 cleaves intact bL31 and generates short bL31 during ribosome preparation from wild-type cells. Intact bL31 was required for subunit association, and its eight cleaved C-terminal amino acids contributed to this function. 70S ribosomes protected bL31 from cleavage by protease 7, but free 50S did not. In vitro translation was assayed using three systems. The translational activities of wild-type and Δ*rpmE* ribosomes were 20% and 40% lower than those of Δ*ompT* ribosomes, which contained one copy of intact bL31. The deletion of bL31 reduces cell growth. A structural analysis predicted that bL31 spans the 30S and 50S subunits, consistent with its functions in 70S association and translation. It is important to re-analyze in vitro translation with ribosomes containing only intact bL31.

## 1. Introduction

### 1.1. Separation of Ribosomal Proteins (r-Proteins) Using 2D PAGE

In 1953, Watson and Crick successfully analyzed the structure of DNA. Using *Escherichia coli*, several researchers subsequently studied the mechanism by which the three sequence codons of the four types of bases that constitute mRNA are translated into 20 amino acids. Translation occurs on ribosomes. In *E. coli*, the 70S ribosome consists of two subunits, 30S and 50S, and is a complex of three types of RNA and more than 50 proteins. The identification of these proteins was a major challenge. In 1970, Kaltschmidt and Wittmann separated r-proteins using two-dimensional gel electrophoresis (KW method), which has an outstandingly high resolution, and the 30S and 50S proteins were named S1–S21 and L1–L34, respectively. The 50S protein L31 was also found as a small spot in the weakly basic region of the two-dimensional gel generated using the KW method (Appendix A) [1]. Subsequently, L7 was found to be N-terminal acetylated L12, and L8 was found to be a complex of L7, L12, and L10; thus, both L7 and L8 were removed from the nomenclature. L26 was found to be identical to S20. Thus, 52 r-proteins were identified. Although the definition of r-proteins seemed to be complete, the two-dimensional pattern generated using the KW method showed many protein-derived noises in addition to the original r-protein spots. Furthermore, the r-protein spots were unevenly sized, suggesting that the noise originated from r-proteins.

### 1.2. Improvement of the 2D PAGE Method

After repeated trial and error to clarify the cause of the noise in the KW method, we found that the main cause is a single amino acid residue, namely, cysteine, which is a constituent of proteins. Under oxidative conditions, in proteins with several cysteine residues, the SH groups on the side chains of two cysteine residues are oxidized and protons are released from the SH groups, forming a loop-like structure with S-S cross-links between the side chains. In addition, S-S cross-links form between cysteine residues of two protein molecules. Consequently, intra- and inter-molecular S-S cross-links change the movement of proteins, generating a large amount of noise in addition to the original r-protein spots. This cannot be resolved simply by reducing the protein sample before electrophoresis. To convert the entire gel from an oxidative to a reducing environment, we pre-treated gels with an electrically charged reducing agent (mercaptoethylamine HCl) prior to the application of the protein molecules, thereby converting the gel from an oxidative to a reducing state, which efficiently prevented protein molecules from being oxidized during migration. The second source of the noise was the gelation of the gel material monoacrylamide. The free radicals necessary for gelation remained in the gel, and proteins underwent various chemical changes. To prevent these changes, a sample-charging gel was introduced before first-dimension electrophoresis, proteins were concentrated in the gel, and then gel pieces containing the protein bands were excised and inserted into the first-dimension gel. Mercaptoethylamine HCl, which was used for reduction, also worked well as a radical scavenger. The third problem was the pH of the second dimension. The pH of the second dimension in the KW method is pH 4.5; therefore, proteins that are highly basic and have small molecular weights catch up with the two-dimensional front line and cannot form a spot. As a countermeasure, the pH of the second dimension was lowered from 4.5 to 3.6 to accelerate the front line such that it outpaced all proteins. This means that all proteins have lower mobility than the front line and form normal spots.

In 1986, this improved method was named radical-free and highly reducing (RFHR) 2D PAGE (Figure 1) [2,3]. The noise disappeared completely, and protein molecules dispersed in the noise converged to the original r-protein spots. With this improvement, r-proteins L35 and L36, which could not be detected using the KW method, were found. L35 contains a single cysteine residue, which is oxidized during migration, causing intermolecular S-S cross-linking; therefore, L35 does not reach the original spot in the KW method. L36 has a small molecular weight, high basicity, and high two-dimensional mobility. Therefore, it is trapped in the front line at pH 4.5 in the second dimension of the KW method and forms a linear pattern on the front line; however, with the adjustment of the pH to 3.6, the original spot was formed and the identification became possible. Consequently, two additional *E. coli* r-proteins were identified using the RFHR 2D PAGE method, bringing the total number of *E. coli* r-proteins to 54, all of which were identified (Figure 1a,c) [2,3]. In summary, there are 21 types of 30S protein (S1–S21) and 34 types of 50S protein (L1–L36, excluding L7 and L8), with the exception of L26, which is identical to S20, yielding a total of 54 types.

## 2. Characterization of bL131 and Its Truncated Form

### 2.1. Detection of Two Spots Corresponding to bL31

In the KW method, bL31 was detected as a single small spot (Appendix A). In the RFHR method, the copy number of bL31 was also low, but a new spot was found in the vicinity of the conventional spot (Figure 1a). It was easily deduced that this spot was derived from bL31 by comparing the position of the spot in the oxidative KW method with that in the reductive RFHR method (Figure 1a–c). Proteins with several cysteine residues are always highly mobile in both the first and second dimensions. bL31 contains four cysteine residues (Figure 2), and thus, forms a loop structure in the oxidative KW method, which increases its mobility. Figure 1 compares the 2D PAGE patterns in reductive and oxidative environments using the apparatus of the RFHR method. Comparing the two spots of bL31 in Figure 1a,b, the reduction shown in Figure 1a slowed the migration of the two spots in both the first and second dimensions, and both spots shifted to the upper left. The new bL31 spot was assumed to have a larger molecular weight and a more basic isoelectric point than the conventional bL31 spot. The new spot found using the RFHR method was named C or precursor bL31 [6].

The amino acid sequence of the protein in spot C matched that of bL31 encoded by *rpmE* [7]. On the other hand, we found that the protease-7-defective AD202 strain has one copy of the new spot, but does not have the conventional bL31, suggesting that the conventional bL31 may be an artifact caused by protease 7; therefore, the new spot was named intact bL31 [8]. Conventional bL31 was defined as short bL31, which lacked eight amino acids at the C-terminus (Figure 2). Intact bL31 was also observed by other groups [9,10]. There was speculation that short bL31 might be produced by a significant physiological modification after the expression of intact bL31, but this remains unknown [10].

### 2.2. Intact bL31 Is Cleaved by Protease 7

To examine how short bL31 is produced from intact bL31, we prepared r-proteins from an *E. coli* wild-type strain and an *ompT*-deficient (∆*ompT*) strain, which lacks the outer membrane protein protease 7 (accession ID:EG10673 (EcoCyc)) encoded by *ompT*, and analyzed them using the RFHR method. In the wild-type strain, intact and short bL31s both had low copy numbers (Figure 3a), while the ∆*ompT* strain had nearly one copy of intact bL31 and no short bL31 (Figure 3b). How do these differences arise? In the wild-type strain, when the outer membrane is ruptured to prepare ribosomes, intracellular ribosomes may encounter outer-membrane-bound protease 7, which should not occur in living cells. Protease 7 has enzymatic activity, hydrolyzing adjacent basic amino acids. The residue at position 62 at the C-terminus of short bL31 is K and the residue at position 63 of intact bL31 is R; therefore, it is possible that protease 7 cleaves between 62K and 63R to generate short bL31. The detection of short bL31 in preparations of ribosomes from wild-type cells strongly suggests that ribosomes and protease 7 actually encounter each other. Recent cross-linking/mass spectrometry in wild-type cells also revealed that L31 exists in a non-cleaved state. Furthermore, this was confirmed by the identification of the C-terminal peptide of full-length L31 during peptide mass fingerprint analysis [11]. Among r-proteins, only bL31 is cleaved by protease 7. In addition, it was reported that initiation factors IF3 and IF2 are also cleaved by protease 7 during cell disruption [12,13,14].

### 2.3. Localization of bL31 on the Ribosome

Twelve inter-subunit bridges connecting the 50S and 30S subunits were reported: B1a/b, B2a/b/c, B3, B4, B5, B6, B7a/b, and B8. Five of these bridges are between 16S rRNA and 23S rRNA. In addition, three are between 30S proteins (bS13, bS15, and bS19) and 23S rRNA, and three more are between 50S proteins (bL2, bL14, and bL19) and 16S rRNA [15]. B1b is a unique bridge that is formed mainly by the single protein molecule bL31 bridging the 50S and 30S subunits. The N-domain of bL31 binds to the central protuberance of the 50S subunit and interacts with the 5S rRNA and bL5 protein. It also interacts with bS13 at the 30S and 50S contact surface [16]. On the other hand, the C-domain binds to the 30S head domain, interacts with bS13 and bS19, and then extends to bS14, where its C-terminus binds to helix 41 of 16S rRNA (Appendix A) [17,18,19,20]. Recent studies using protein-to-protein cross-linkers revealed the interactions of bL31–bL5, bL31–bS19, and bL31–bS14. The contribution of cross-linking by a single molecule to the association of both subunits appears to be stronger than that of other bridges [11]. It is known that bridge B1b facilitates the initiation of protein synthesis and the maintenance of translational fidelity [21].

Another protein inter-bridging subunit might exist. Although bS20 and bL26 are identical, they are detected in both subunits when the 70S ribosome is dissociated into the 50S and 30S subunits, with bS20 in 30S and bL26 in 50S. Therefore, the single molecule bS20/bL26 may be bound across both subunits, similar to bL31. However, the copy number of bS20/bL26 in the 70S ribosome is 1.38, which is higher than 1 [2]; therefore, we cannot rule out the possibility that it binds to each subunit independently.

### 2.4. The Effect of Protease 7 on the Subunit Association of Ribosomes Prepared Using the Conventional Extraction Method

The association and stability of the 70S particle in vitro strongly depend on the concentrations of various components, including ribosomes, Mg^2+^, mono-cations, and polyamines [22,23,24,25]. To compare the association capacity of ribosomes, we simply exploited the Mg^2+^ dependency of association. We examined the effect of protease 7 on ribosomes prepared via the conventional extraction method using quartz sand crushing. The *E. coli* W3110 wild-type strain, the ∆*ompT* strain lacking protease 7, and the ∆*rpmE* strain lacking bL31 were used to study the effect of bL31 shortening on the structure and function of ribosomes. The buffer contained 100 mM NH_4_ acetate; 20 mM Tris-HCl (pH 7.6); and 2, 5, 8, 10, or 15 mM Mg^2+^ acetate. Cells were harvested in the exponential growth phase, and high-salt-washed ribosomes (HSRs) were prepared from crude ribosomes (CRs) and analyzed by means of 5–20% sucrose density gradient (SDG) centrifugation. In the buffer containing 8–15 mM Mg^2+^, all ribosomes were 70S particles. In the buffer containing 5 mM Mg^2+^, almost all ribosomes were 70S particles in the ∆*ompT* strain, but almost all 70S ribosomes dissociated into 50S and 30S in the ∆*rpmE* strain. In the wild type, 70S decreased and the dissociated 50S and 30S appeared, reflecting the damage caused by protease 7. In a buffer containing 2 mM Mg^2+^, all ribosomes in all strains dissociated into 50S and 30S (Figure 4).

Proteins (HSRs) were then extracted from cells in the exponential growth phase and separated using the RFHR 2D PAGE method (Figure 5a). The protein levels in the intact and short bL31 spots were measured using densitometry (Figure 5b). The copy number of intact bL31 was 0.81 for non-incubated ∆*ompT* cells but was decreased to 0.31, 0.19, and 0.12 for wild-type cells that were not incubated, incubated at 4 °C for 30 min, and incubated at 37 °C for 30 min, respectively. Conversely, the copy number of short bL31 increased to 0.43, 0.54, and 0.69 for wild-type cells that were not incubated, incubated at 4 °C for 30 min, and incubated at 37 °C for 30 min, respectively, which reflected the increase in free 50S (Figure 5a,b) [26].

### 2.5. The Protein Levels of Intact and Short bL31s in the 70S and 50S Fractions

Next, we attempted to compare the protein levels of intact and short bL31s in the 70S and 50S fractions. Cells of the wild-type strain in the log phase were crushed in the usual way and extracted with the usual 15 mM Mg^2+^-containing buffer to obtain CRs. The CRs were subjected to preparative SDG centrifugation with 15 mM Mg^2+^-containing buffer to obtain the 70S and 50S fractions (Appendix A). Both fractions were analyzed by means of the RFHR 2D PAGE method (Appendix A). The 50S fraction mostly contained short bL31, while the 70S fraction contained both intact and short bL31s. It is possible that some of the 70S particles in this experiment were free 50S with short bL31 that was reassociated with 30S in the 15 mM Mg^2+^-containing buffer.

To confirm this reassociation, cells of the wild-type strain in the exponential growth phase were crushed in the usual way and extracted with 5 mM Mg^2+^-containing buffer to obtain CRs. The CRs were subjected to preparative SDG centrifugation with 5 mM Mg^2+^-containing buffer to obtain the 70S and 50S fractions. The 70S fraction contained only intact bL31, while the 50S fraction contained only short bL31 (Figure 6a). It is thought that the reassociation of free 50S and 30S in 15 mM Mg^2+^-containing buffer was prevented in 5 mM Mg^2+^-containing buffer. This indicates that only intact bL31 binds to the 70S fraction and only short bL31 binds to the free 50S fraction.

To further correct for the intact/short bL31 bias, we used ∆*ompT* cells and generated 50S subunits and 70S particles with only intact bL31 from these cells. Ribosomes were obtained from ∆*ompT* cells in the log phase, and 50S and 70S fractions with only intact bL31 were acquired (Figure 6b upper). These fractions were mixed with CD obtained by means of induction from a W3110 derivative harboring an IPTG-induced *ompT* overexpression plasmid encoding protease 7. After incubation at 37 °C for 30 min, we analyzed the proteins in the mixtures using the RFHR method. The 70S ribosomes contained approximately one copy each of intact bL31 and trace levels of short bL31. However, the 50S subunits contained large amounts of short bL31 and virtually no intact bL31, in stark contrast with the samples not treated with CD (Figure 6b). These results suggest that intact bL31 is stable when bound to 70S ribosomes, but is highly sensitive to protease 7 when bound to free 50S subunits, indicating that intact bL31 of 70S ribosomes is protected from protease 7 by 30S.

When wild-type cells are disrupted, the ribosomes they contain should be in various steps of the ribosome cycle. Ribosomes in the elongation step of the ribosome cycle are in an associated state, while those in the initiation and recycling steps are dissociated into free subunits. After cell disruption, protease 7 bound to the outer membrane encounters free 50S or associated 70S. It cleaves intact bL31 to produce short bL31 when it encounters free 50S but is prevented from cleaving intact bL31 using 30S when it encounters 70S. The ribosomes prepared should therefore be a mixture of 70S with intact bL31 and free 50S with short bL31. Naturally, the content of ribosomes will vary depending on the cell disruption method, amount of time, and temperature used to prepare them. The ribosome cycle will continue even in disrupted cells; therefore, all free 50S will be attacked by protease 7 and the amount of short bL31 will increase over time.

### 2.6. The Effect of bL31 on the Translational Activity

The effect of bL31 on in vitro translational activity was examined in three ways. First, to compare HSRs prepared from the ∆*ompT* and ∆*rpmE* strains, we measured the in vitro synthesis of the dihydrofolate reductase (DHFR) (1–152) protein using the transcription/translation PURE system [27]. The amount of protein synthesized in the ∆*rpmE* strain was 0.62 relative to that in the ∆*ompT* strain (set to 1). This indicates that translational activity is reduced by about 40% in the absence of bL31 (Figure 7a).

Next, poly-U-dependent poly-phenylalanine incorporation [28] was measured using HSRs prepared from the ∆*ompT* and wild-type strains. In this system, 70S binds to poly-U mRNA without dissociating into the 50S and 30S subunits. Elongation is initiated via the direct binding of phenylalanine-tRNA to P and A sites. Phenylalanine incorporation was 0.80 in the wild-type strain relative to that in the ∆*ompT* strain (set to 1). There is no normal initiation step in this system; therefore, this decrease in activity is thought to be due to a difference in the elongation step (Figure 7b).

Next, we measured MS2 mRNA-dependent leucine incorporation [28] using HSRs prepared from the ∆*ompT* and wild-type strains. In this system, the Shine Dalgarno sequence, an initiation codon, initiation factors, and methionine-tRNA are required, and leucine incorporation is initiated via a natural translational process. Leucine incorporation was about 0.78 in the wild-type strain relative to that in the ∆*ompT* strain (set to 1), which was almost equal to the decrease in activity observed in the poly(U) system (Figure 7c). Therefore, the initiation step does not seem to markedly differ between the *∆ompT* and wild-type strains, and the elongation step, which is common to both the poly(U) and MS2 systems, is considered to be perturbed by the shortening of bL31.

The results of the three assays described above are consistent. The 40% decrease in translation activity in the ∆*rpmE* strain in the PURE system is due to the absence of bL31 on the ribosome. On the other hand, the copy number of short bL31 in the non-incubated wild-type strain was 0.43 (Figure 5b). However, some short bL31 is released from ribosomes; therefore, about half of bL31 is short if short bL31 detached from ribosomes is considered.

Translational activity in the poly(U) and MS2 systems is about 20% lower in the wild-type strain than in the ∆*ompT* strain; therefore, the reduction in activity due to the shortening of bL31 is about 40%, consistent with the 40% reduction in the ∆*rpmE* strain. This means that ribosomal damage caused by the shortening of bL31 is equivalent to that caused by the loss of bL31.

### 2.7. bL31 Is Involved in Hibernation

In 1990, we first discovered the 100S ribosome in *E. coli* during the stationary phase [6]. Two 100S-forming protein factors, designated as RMF (55 amino acids) and HPF (95 amino acids), are expressed almost simultaneously in the stationary phase and bind to ribosomes [8,29,30,31,32,33,34]. 100S is a dimer of 70S with the form 50S/30S–30S/50S [8,31,35,36]. Cells survive in a stressed environment. This activity was named ribosome hibernation [31]. Although RMF- and HPF-mediated 100S formation is specific to the Proteobacteria gamma group, we subsequently found that a different mechanism of 100S formation exists in all bacteria [37]. In this case, 100S formation occurs through the dimerization of long HPF (185 amino acids) alone as a hibernation factor, which binds to 70S ribosomes [38,39,40].

We examined the contribution of bL31 to 100S formation by RMF and HPF in *E. coli* by comparing CRs prepared from cells (3-day culture) of the ∆*ompT* strain containing only intact bL31, the wild-type strain containing both intact and short bL31s, and the ∆*rpmE* strain lacking bL31 via SDG centrifugation. The ∆*ompT* and wild-type strains formed 100S, which outperformed 70S, but the *∆rpmE* strain formed only a few 100S ribosomes. RMF expression levels were the same in the three strains (Appendix A). These results indicate that bL31 is involved in 100S formation.

### 2.8. YkgM, Which Is a bL31 Paralog, Maintains Translation Activity by Replacing bL31 When It Defects

An important function of bL31 in response to environmental changes is to supply cells with Zn^2+^. bL31, together with bL36, is a Zn^2+^-binding r-protein [41,42,43,44]. The paralogs of bL31 and bL36, namely, YkgM and YkgO, are encoded in the YkgM operon and negatively regulated by the ZUR repressor. When the cellular Zn^2+^ concentration decreases, the ZUR repressor of Zn^2+^-binding proteins is inactive, and *ykgM* and *ykgO* are expressed [45,46]. When bL31 and bL36 dissociate from the ribosome, YkgM and YkgO bind to the binding sites of bL31 and bL36, respectively. bL31 and bL36 detached from the ribosome release Zn^2+^, and released zinc atoms are used by other cellular zinc-binding proteins, such as DNA polymerase and primase [42,47,48,49,50].

The ∆*rpmE* strain grew less well than the wild-type strain and formed small colonies. We isolated five mutant strains that grow as well as the wild-type strain and form normal colonies. All five strains had independent mutations of the *zur* gene and inactivated ZUR repressor [51]. In the exponential growth and stationary phase, the copy number of YkgM or YkgO in ribosomes of the ∆*rpmE* + *zur::IS2* mutant strain for (a) or ∆*zur* ∆*rpmJ ompT*::*Km* for (b) was 1 (Appendix A), and the in vitro translation activity in this strain was comparable with that in the W3110 ∆*ompT:*:*Km* and W3110 ∆*rpmE*::*Km*, *zur*:*:IS2* with YkgM strains (Appendix A). When YkgM and bL31 co-existed in cells, YkgM bound more strongly to ribosomes than bL31 (Appendix A). This indicates that the translation activity required for growth is maintained in the *zur* mutants due to the replacement of bL31 by YkgM in ribosomes.

YkgM and YkgO lack a protease 7 cleavage site and are therefore not attacked by protease 7 during ribosome preparation [51].

### 2.9. The Effect of bL31 Function on Bacterial Growth

Finally, the effect of bL31 on bacterial growth was examined by culturing the W3110 wild-type, ∆*ompT*, and ∆*rpmE* strains in an EP medium [2]. The ∆*rpmE* strain had a doubling time of 35–40 min in the exponential growth phase, was short-lived in the stationary phase (Appendix A), and formed small colonies, while the W3110 wild-type and ∆*ompT* strains had a doubling time of 20–25 min. This suggests that bL31 is required for normal growth of *E. coli* and in vitro translation activity.

## 3. Conclusions

Short bL31 is produced via the cleavage of intact bL31 between 62K and 63R. We found that this cleavage is mediated by protease 7 bound to the outer membrane during ribosome preparation. This artificial accident has gone unnoticed for a long time.

We examined how the shortening of bL31 affects the structure and function of ribosomes. The effect of short bL31 on subunit association was examined by comparing wild-type, ∆*ompT*, and ∆*rpmE* strains using SDG centrifugation. Wild-type cells displayed an increase in subunit dissociation in proportion to the percentage of short bL31 in ribosomes, indicating that bL31 shortening has a damaging effect equivalent to that of bL31 loss in ∆*rpmE*. Structural studies centered on cryo-electron microscopy showed that bL31 is a very unique protein that bridges the 50S and 30S subunits. The cleavage site to generate short bL31 (62K–63R) is located near the bS14 of 30S. bL31 bound to 70S during cell disruption is protected from protease 7-mediated cleavage at this site by 30S, whereas bL31 bound to free 50S is cleaved by protease 7 due to the lack of protection by 30S.

We examined the effect of short bL31 on in vitro translation. Translation activity was about 40% lower in ∆*rpmE* cells, which lack bL31, than in ∆*ompT* cells, which have one copy of intact bL31. On the other hand, the translation activity was about 20% lower in wild-type cells than in ∆*ompT* cells. If we assume that bL31 shortening occurs in about half of the ribosomes in the wild-type strain and that the remaining bL31 is intact, the damage caused by bL31 shortening in wild-type cells corresponds exactly to the damage caused by bL31 loss in ∆*rpmE* cells. This damage is suggested to occur at the elongation stage, during which ribosomes function in the 70S state. However, the effect of short bL31 in in vitro assays may extend beyond elongation to other stages of the ribosome cycle. In particular, during the initiation and recycling phases of the ribosome cycle, ribosomes function as free 50S and 30S. In vitro assays of the initiation and recycling phases of the ribosome cycle in wild-type strains likely include free 50S with short bL31 produced by protease 7 in addition to the original free subunit with intact bL31.

It is important to perform in vitro assays of the entire ribosome cycle using ribosomes containing only intact bL31 to correctly establish the ribosome cycle.

## Figures and Tables

**Figure 1 ijms-24-03445-f001:**
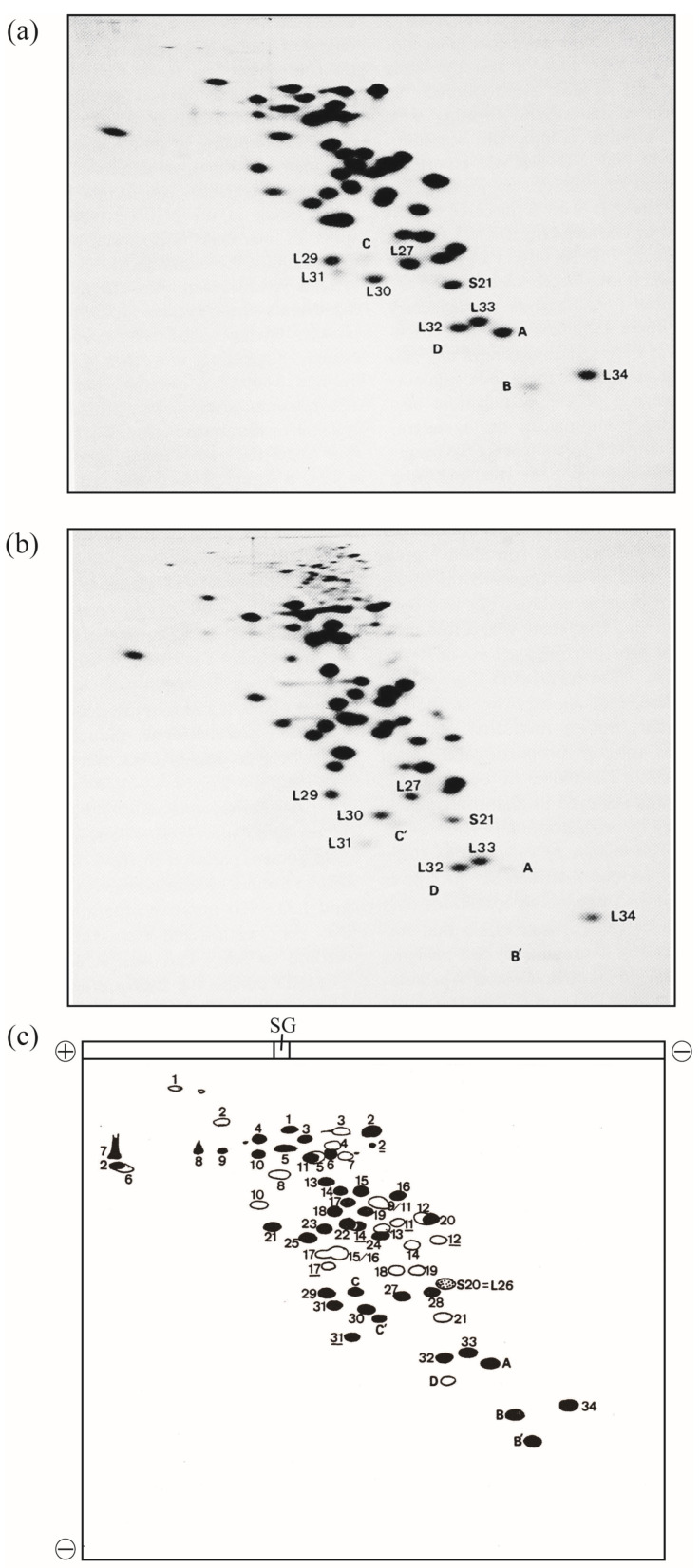
Electrophoretograms of r-proteins prepared from high-salt-washed 70S ribosomes of the *E. coli* strain Q13. *E. coli* r-proteins were prepared using the previously described acetic acid method [4]. A one-tenth volume of 1 M MgCl_2_ and two volumes of acetic acid were added to the ribosomal solution, and the mixture was stirred at 0 °C for 1 h. After centrifugation at 10,000 g for 10 min at 4 °C, the supernatant was dialyzed three times against 2% acetic acid (the volume of the dialysis buffer was 300-fold larger than the volume of the sample) for 24 h. The proteins were lyophilized and stored at −80 °C until use. The protein solution (2 mg of protein in 100 µL of 8 M urea containing 0.2 M 2-mercaptoethanol) was analyzed using RFHR 2D PAGE, as described previously [2,3], with slight modifications [5]. Sample charging electrophoresis was performed at 100 constant volts (CV) for 15 min at room temperature (RT). Subsequently, the first-dimensional electrophoresis was performed at 170 CV for 8 h at RT, and electrophoresis in the second dimension was performed at 100 CV for 15 h at RT. The 2D gels were stained with CBB G-250, and protein spots were scanned using a GS-800 calibrated densitometer (Bio-Rad Laboratories Inc., Hercules, CA, USA). (**a**) Under reducing conditions, electrophoresis was performed as described previously [2]. The amount of r-proteins was 0.6 mg/gel. (**b**) Under non-reducing conditions, electrophoresis was performed as in (**a**), except thiol reagents were not used and incubation prior to sample charging was not performed. Spots of new proteins (A, B, C, D, B′, C′) and vicinal spots of known r-proteins are shown. (**c**) Scheme of 2D electrophoretograms (**a**,**b**). Solid spots are those of L-proteins or proteins found in 50S subunits. Open spots are those of S-proteins or proteins found in 30S subunits. L and S are omitted from their names, except for S20 = L26 (dotted spot). Spots formed by non-reduced r-proteins containing multiple cysteine residues are underlined. The spot shifts of L2, L14, L31, B, and C upon reduction are shown in the accompanying paper. Sample gel is abbreviated to SG. The current directions in one- and two-dimensions are shown by plus and minus signs. L31: short L31, C&C′: intact L31, A: L35, B&B′: L36, and D: SRA (S22). In Figure 1, the “b” of r-proteins is omitted.

**Figure 2 ijms-24-03445-f002:**
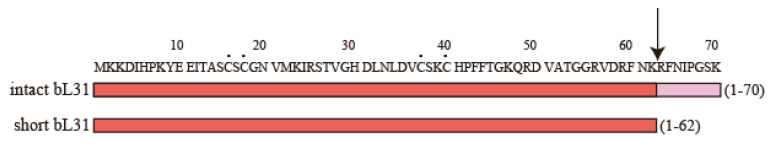
Amino acid sequences of intact and short bL31s. The amino acid sequence of intact bL31 (70 amino acids) (P0A7M9 · RL31_ECOLI, Uni Prot) is shown at the top, with amino acids 1–62 and 63–70 shown in red and pink, respectively. Short bL31 (62 amino acids), shown in red at the bottom, is produced via the cleavage of intact bL31 by protease 7. The arrow and black dots indicate the site cleaved by protease 7 and cysteine residues, respectively. In Figure 2, the “b” of r-proteins is omitted.

**Figure 3 ijms-24-03445-f003:**
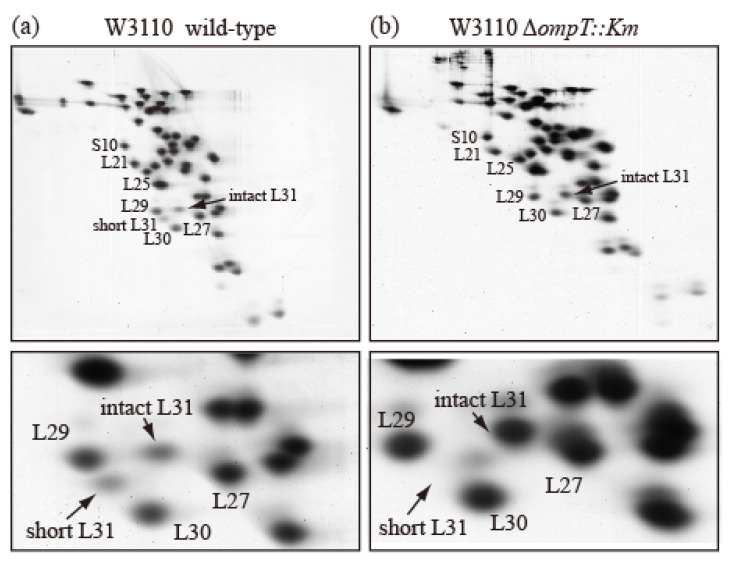
Ribosomes prepared from wild-type *E. coli* cells contain both intact and short bL31s. (**a**) W3110 wild-type and (**b**) W3110 Δ*ompT::Km* cells were grown at 37 °C in EP medium and collected in the exponential growth phase (Klett units: 50), and HSRs (high-salt-washed ribosomes) were prepared. HSR proteins were analyzed using RFHR 2D PAGE. Gels were stained with CBB. (**Upper**) Spots corresponding to r-proteins bS10, bL21, bL25, bL27, bL29, bL30, intact bL31, and short bL31 are indicated. (**Lower**) An enlarged image of the bL31 spot and surrounding area is shown. In Figure 3, the “b” of r-proteins is omitted.

**Figure 4 ijms-24-03445-f004:**
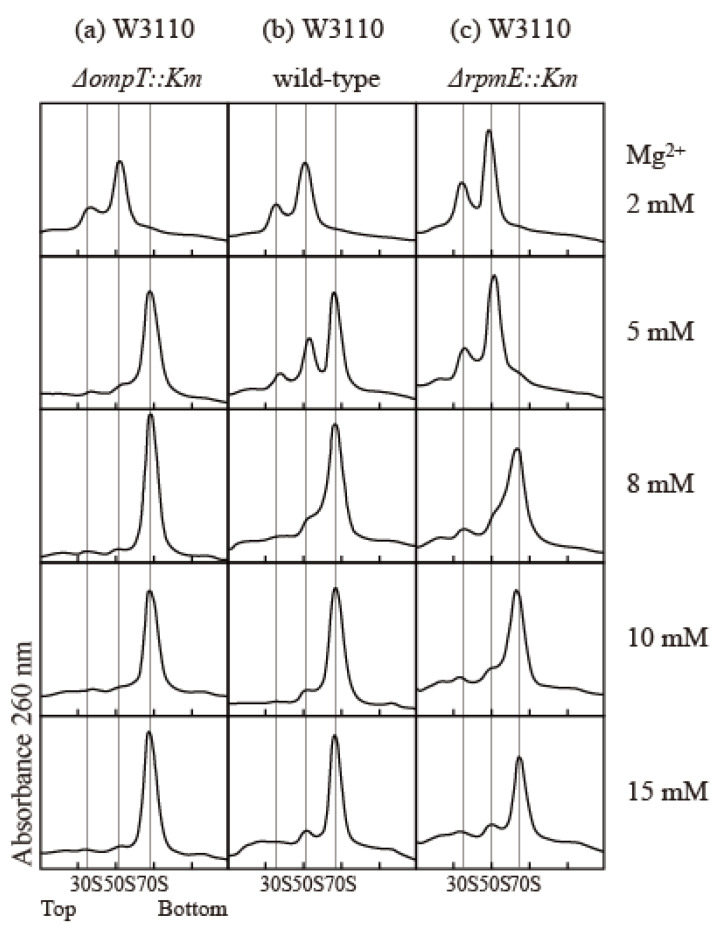
Influence of bL31 on ribosomal profiles upon SDG centrifugation. HSRs of W3110 *ΔompT::Km*, wild-type, and Δ*rpmE::Km* cells were prepared and analyzed using 5–20% SDG centrifugation with the indicated Mg^2+^ concentrations. The resultant ribosome profiles are shown. (**a**) W3110 *ΔompT::Km*, (**b**) W3110 wild-type, and (**c**) W3110 *ΔrpmE::Km*. HSRs (60 pmol) were subjected to SDG centrifugation.

**Figure 5 ijms-24-03445-f005:**
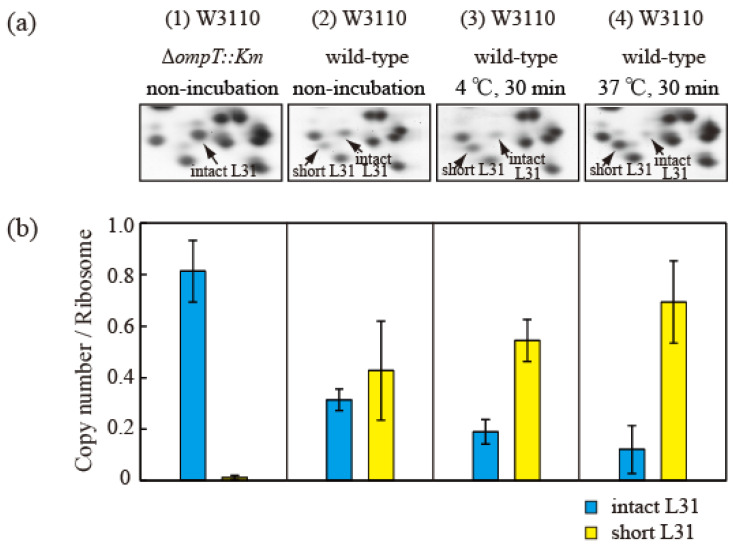
Intact bL31 is cleaved by protease 7 to generate short bL31 during the incubation of disrupted cells. W3110 Δ*ompT::Km* and wild-type cells were grown at 37 °C in EP medium and collected in the exponential growth phase (Klett units: 50), and high-salt-washed ribosomes (HSRs) were prepared. These samples were designated (1) and (2) “non-incubation”. To stimulate cleavage of the C-terminal residues of intact bL31, disrupted W3110 cells were incubated at (3) 4 °C or (4) 37 °C for 30 min. (**a**) r-proteins of the four samples of HSRs were analyzed by means of RFHR 2D PAGE. Relevant gel areas are shown. Arrows indicate spots corresponding to intact and short bL31s. (**b**) The protein copy number per ribosome was determined. Copy numbers of intact and short bL31s are shown as cyan and yellow bars, respectively. The experiment was repeated at least three times. The bars and error bars show the mean copy numbers of bL31 and standard deviations, respectively. In Figure 5, the “b” of r-proteins has been omitted.

**Figure 6 ijms-24-03445-f006:**
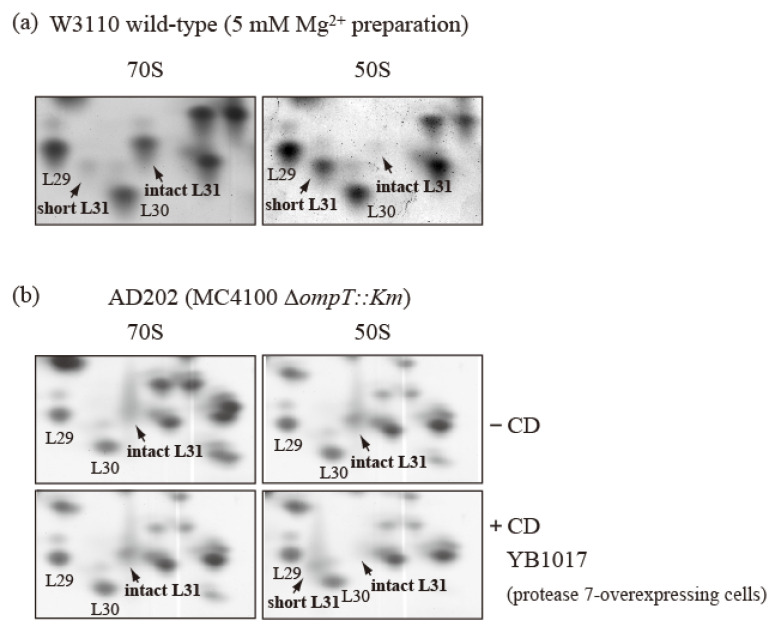
bL31 is accessible for cleavage by protease 7 when it is associated with the free 50S subunit but not when it is associated with the 70S ribosome. (**a**) 70S and 50S of W3110 cells prepared using a buffer containing 5 mM Mg^2+^ contain only intact and short bL31s, respectively. W3110 cells were grown at 37 °C in EP medium and collected in the exponential growth phase (Klett units: 50). CRs were prepared using the standard methods, except that buffer I was modified to contain 5 mM Mg^2+^. CRs were fractionated via 10–40% SDG centrifugation in the presence of 5 mM Mg^2+^, and 50S and 70S fractions were collected. Each fraction was analyzed using RFHR 2D PAGE. Relevant gel areas of bL31 are shown. (**b**) bL31 is more accessible for cleavage by protease 7 in free 50S than in 70S. Δ*ompT::Km* (AD202) cells were grown at 37 °C in EP medium and collected in the exponential growth phase (Klett units: 50), and HSRs were prepared as the 70S fraction. HSRs were dialyzed against dissociation buffer I (20 mM Tris-HCl (pH 7.6), 1 mM magnesium acetate, 100 mM ammonium acetate, and 6 mM 2-mercaptoethanol) overnight. The sample was fractionated by means of 10–40% SDG centrifugation in dissociation buffer I, and the 50S fraction was collected. The 50S and 70S fractions exclusively contained intact bL31. CD was prepared from protease-7-overproducing W3110 (YB1017) cells and added to the 50S and 70S fractions. Each mixture (50S ± CD and 70S ± CD) was incubated at 37 °C for 30 min. Subsequently, proteins in the four samples were analyzed using RFHR 2D PAGE. Relevant gel areas of bL31 are shown. Spots corresponding to r-proteins bL29, bL30, intact bL31, and short bL31 are indicated. In Figure 6, the “b” of r-proteins is omitted.

**Figure 7 ijms-24-03445-f007:**
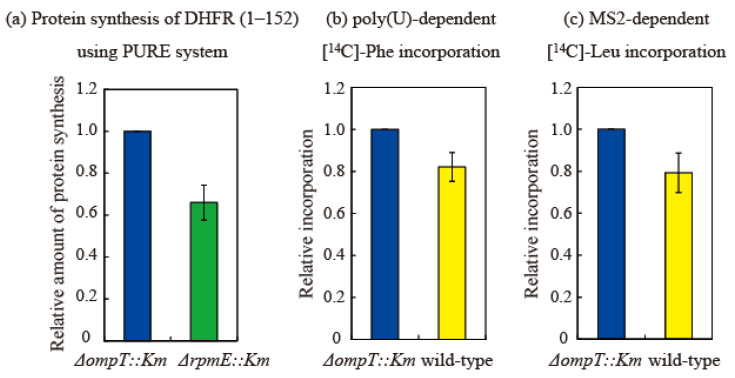
Translational activity of ribosomes containing short bL31 is lower than that of ribosomes containing intact bL31. (**a**) Synthesis of DHFR (1–152) is 40% lower in Δ*rpmE::Km* HSRs than in Δ*ompT::Km* HSRs. Synthesis of DHFR (1–152) using the PURE system [27] was performed in vitro using HSRs containing exclusively intact bL31 (from Δ*ompT::Km* cells) or no bL31 (from Δ*rpmE::Km* cells). After the electrophoresis of reaction mixtures on 10–20% SDS-PAGE gels, the synthesized DHFR (1–152) was visualized using CBB. The density of the DHFR (1–152) band was normalized against that of the r-protein band. Protein synthesis was normalized against synthesis using HSRs containing only intact bL31 (i.e., Δ*ompT::Km*). The data are averages of four independent pure system reactions and two PAGE gels per reaction (a total of eight gels). (**b**,**c**) Translational activity in vitro is 20% lower in wild-type HSRs than in Δ*ompT::Km* HSRs. (**b**) Poly(U)-dependent L-[U-^14^C] phenylalanine incorporation and (**c**) MS2-dependent L-[U-^14^C] leucine incorporation were performed in vitro using HSRs containing only intact bL31 (Δ*ompT::Km*) or both intact and short bL31s (wild-type). Protein synthesis was normalized against synthesis using HSRs containing only intact bL31 (i.e., Δ*ompT::Km*). Net incorporation of L-[U-^14^C] Phe (**b**) and L-[U-^14^C] Leu (**c**) using control Δ*ompT::Km* HSRs was equivalent to 35.6 and 1.0 nmol, respectively. Relative amounts of protein synthesis using HSRs of Δ*ompT::Km,* Δ*rpmE::Km*, and the wild type are shown by the midnight blue, green, and yellow bars, respectively. The data represent the averages of three independent experiments.

## Data Availability

All of the data in this review, except for one case, are included in the published articles [26,51].

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
