# Peer review of "The Discovery of Ribosomal Protein bL31 from Escherichia coli: A Long Story Revisited"

_ijms, 2023, doi:10.3390/ijms24043445_

Round 1

Author Response

According to comment, influences of protease 7 to IF3 and IF2 were added to Results and Discussion 2-2 with references [11-13].

Figure 5 and 7 have been modified for clarity.

Reviewer 2 Report

The manuscript "Cleavage by protease 7 abrogates the functions of ribosomal 2 protein L31 in subunit association and translation in Escherichia coli" by Akira Wada and co-authors describes experiments aiming at the understanding of the physiological role of L31 proteolytic processing. Authors demonstrated that L31 protein is migrating as two separate spots on a specific 2D gel electrophoresis. Authors claim the spots differ by a cleavage of C-terminal 8 aminoacids. These aminoacids were suggested to be cleaved with OmpT outer membrane protease. Authors demonstrate that cleavage reduces ribosomal subunit association and efficiency of protein synthesis. 

The study is thoroughly done and the manuscript provides a reader with a new information which is of interest to scientists working in the field of ribosome structure and function. At the same time, there several issues that might be resolved to improve the study.

1. Spots corresponding to L31 have to be identified by mass-spectrometry to support the claims. Same applies to the identification of an exact cleavage site. 

2. I would recommend to model the situation when cleaved L31 is solely present in a cell.  To this end rpmE knockout might be complemented by a plasmid coding for an intact of truncated L31. These strains might be tested for translation efficiency and subunit association.

3.  OmpT is an outer membrane protease. Ribosomes, to my knowledge, function inside the cell. Evidence should be provided that cleavage occur in vivo rather than in the course of cell lysis.

Author Response

  1. Two spots of intact and short bL31 on RFHR 2D PAGE were identified by our mass spectroscopy (MD TOFF MS). The cleavage site was  determined by molecular weight of short bL31 from our mass spectroscopy and  C-terminal amino acids sequence of short bL31 analyzed by Brosius 1978.
  2. Short bL31 is an artifact produced during cell disruption. It does not exist in living cells. Therefore, we have not carried out characterization of short bL31 in living cells.
  3. we demonstrated the absence of short bL31 in living cells in the following experment.

    "We attempted to ascertain whether short L31 is present in living wild-type cells by inactivating protease 7 prior to cell disruption in order to avoid protease 7 and ribosomes encountering each other. In total, 0.5 g of cells of a wild-type strain in log phase was placed in a cooled mortar, and 0.5 mL of 67% acetic acid-0.3 M MgCl2-Tris buffer, which is used to insolubilize nucleic acids and solubilize proteins, was added. The solution was placed at 0℃ for 10 min. The mixture was then gently stirred with a pestle at 4℃. This treatment inactivates protease 7 bound to the outer membrane first, and then disrupts intracellular ribosomal granules and solubilizes all proteins. The mixture was centrifuged at 7000 rpm for 20 min at 4℃, and the solubilized protein fraction was dialyzed in 2% acetic acid. Finally, lyophilized proteins [total proteins excluding cell debris (CD)] were obtained and separated by the RFHR method (Supplementary Figure S2). These results demonstrate that only intact, not short, L31 is present in living wild-type cells. As suggested by analysis of the ∆ompT strain, these data show that cleavage of intact L31 by protease 7 to generate short L31 is not of physiological significance, but an accident caused by outer membrane disruption necessary to prepare ribosomes. Cross-linking/mass spectrometry by other group also revealed that L31 exists in a non-cleaved state. Furthermore, this was confirmed by identification of the C-terminal peptide of full-length L31 during peptide mass finger print analysis.

    The above experiment is an unpublished data, so we do not use in the review.

    Figure 5 and 7 have been modified for clarity.

Reviewer 3 Report

This paper is a total replica of a previous paper published in 2017 by the same group, as shown below. The manuscript uses old technique 2DPAGE and there are no new additional value of the current manuscript when compared to the known facts. Most importantly, there is a serious ethical issue in this paper because most Figures shown have been published previously, and text are highly redundant.

"Masami UetaChieko WadaYoshitaka BesshoMaki MaedaAkira Wada. Ribosomal protein L31 in Escherichia coli contributes to ribosome subunit association and translation, whereas short L31 cleaved by protease 7 reduces both activities. Genes Cells. 2017 May;22(5):452-471. doi: 10.1111/gtc.12488. Epub 2017 Apr 10. PMID: 28397381"

Major:

Figure 1 reiterates a known fact of L31 isoforms that has been reported since 1980-90s. Besides,  the location of L31 has changed and the reason is unclear.

Figure 2 is same as Figure S1A in PMID: 28397381.

Figure  3: The image is exactly the same published as the Figure 1 in the PMID: 28397381.

Figure 4: Apparently, this figure is taken directly from the Figure 4 of the PMID: 28397381. Each figure are similar to those of the column (1), column (2) and column (5).

Figure 6 is exactly the same as the Figure 3 in PMID: 28397381.

Figure 7 is exactly the same as the Figure 7A & 7C in PMID: 28397381. Similar Figure Legend is also detected.

Figure S3 protein structure model is same as Figure S1B in PMID: 28397381.

Figure S5 is same with Figure 8 in PMID: 28397381.

Figure S7B is same with Figure 9B in PMID: 28397381.

I just hope that the authors can be more responsible and ethical.

Author Response

This manuscript is a review, but not an article. Therefore, we used our previous results.

Figure 5 and 7 have been modified for clarity.

At the beginning, I described the progress of the discovery of two bL31s。

Please read it as a review.

Reviewer 4 Report

Manuscript ID: ijms-2145538

Cleavage by protease 7 abrogates the functions of ribosomal protein L31 in subunit association and translation in Escherichia coli

The manuscript describes relevant information about the importance of the ribosomal protein L31 for the translational function of ribosomes and growth in E. coli. Protease 7 is implicated in the release of a short version of L31 ribosomal protein. It was observed a consistent reduction in the translational activity of ribosomes in wild-type and ΔrpmE (L31 deficient) mutant cells with respect to protease 7 deficient cells (ΔompT), thus indicating that cleavage of L31 protein by protease 7 gradually caused a reduction in the ribosomal activity.

The manuscript could be accepted for publication in the International Journal of Molecular Sciences after addressing the following corrections.

Why the template of the manuscript is for “Acoustics”

Include in the abstract of the manuscript some information about the relevance of the L31 in the growth of E. coli.

The name “protease 7” in the manuscript is in two forms “protease 7” and “protease7”, please choose the more adequate and homogenize in the whole manuscript.

Figures quality must be improved, images and font are blurry and have shade

Figure captions are too long and include many methodological aspects, please review them, and adapt them to a more concise form if possible.

In line 140, the period is in red font

In line 353, the period and the reference numbers are underlined

In lines 366 and 367, justify the format of the subtitle

References must not be included in conclusions

Check the alignment or format of the patents information

Author Response

1. Why the template of the manuscript is "Acoustics"

We can't understand “Acoustics”.

2. Include in the abstract of the manuscript some information about the relevance of the L31 in the growth of E. coli.

According to comment, we included the influence of short bL31 to the growth of E. coli cells in abstract.

3. The name “protease 7” in the manuscript is in two forms “protease 7” and “protease7”, please choose the more adequate and homogenize in the whole manuscript.

The name "protease7" was changed to "protease 7".

4. Figures quality must be improved, images and font are blurry and have shade

Unlike in isoelectric point 2D PAGE where a single protein splits into multiple spots, in the RFHR method a single protein converges to a single spot. Therefore, the RFHR method is very effective for quantification of protein amount. In addition, the RFHR method has a high ability to separate r-proteins on the 2D gels.  In this study. quantification of bL31 is very important. We could firmly determine copy number of bL31 by RFHR method.

Figure 5 and 7 have been modified for clarity.

5. Figure captions are too long and include many methodological aspects, please review them, and adapt them to a more concise form if possible.

As we used figure legends in stead of "Materials and Method", the figure legends became unavoidably long to explain experiments.

6. In line 140, the period is in red font

We could not find the red font in line 140.

7. In line 353, the period and the reference numbers are underlined

We removed the underline.

8. In lines 366 and 367, justify the format of the subtitle

According to comment, we justified the format of the subtitle.

9. References must not be included in conclusions

According to comment, we removed references from conclusions.

10. Check the alignment or format of the patents information

These are Japanese patents. We confirmed that these are  official.

Round 2

Reviewer 2 Report

All my concerns were clarified.

Author Response

Thank you very much for your comments.

Reviewer 3 Report

The paper and writing format does not appear as review paper. But rather, a report that includes the results finding. Please stop the plagiarism, and behave like a decent scientist.

Author Response

â—† In this review, we describe the process leading to the discovery of intact bL31 and refinement of 2D PAGE that made it possible

â—† Subsequently, we described that short bL31 is produced by artificial encounter with protease 7.

â—† Next, we described the results of clarifying the properties of short bL31 using the SDG and RFHR methods.

â—† All of the above results have been found by our own experiments, absolutely not plagiarism. (JB, 1986, Genes to Cells, 1998, 2017 and 2020).

â—† We recently performed an experiment to prove the absent of short bL31 in living cells, but it was not included in the review due to unpublished data.

â—† According to editor’s comment, to prevent misunderstanding, subtitle was changed from RESULTS AND DISCUSSION to CHARACTERIZATION OF bL31 AND ITS TRUNCATED FORM.